# Self-Consistent Model for the Compositional Profiles in Vapor–Liquid–Solid III–V Nanowire Heterostructures Based on Group V Interchange

**DOI:** 10.3390/nano14100821

**Published:** 2024-05-07

**Authors:** Vladimir G. Dubrovskii

**Affiliations:** Faculty of Physics, St. Petersburg State University, Universitetskaya Emb. 13B, 199034 St. Petersburg, Russia; dubrovskii@mail.ioffe.ru

**Keywords:** nanowire heterostructures, VLS growth, group V interchange, compositional profiles, modeling

## Abstract

Due to the very efficient relaxation of elastic stress on strain-free sidewalls, III–V nanowires offer almost unlimited possibilities for bandgap engineering in nanowire heterostructures by using material combinations that are attainable in epilayers. However, axial nanowire heterostructures grown using the vapor–liquid–solid method often suffer from the reservoir effect in a catalyst droplet. Control over the interfacial abruptness in nanowire heterostructures based on the group V interchange is more difficult than for group-III-based materials, because the low concentrations of highly volatile group V atoms cannot be measured after or during growth. Here, we develop a self-consistent model for calculations of the coordinate-dependent compositional profiles in the solid and liquid phases during the vapor–liquid–solid growth of the axial nanowire heterostructure $A_{x_{0}}B_{1-x_{0}}C/A_{x_{1}}B_{1-x_{1}}C$ with any stationary compositions $x_{0}$ and $x_{1}$. The only assumption of the model is that the growth rates of both binaries AC and BC are proportional to the concentrations of group V atoms A and B in a catalyst droplet, requiring high enough supersaturations in liquid phase. The model contains a minimum number of parameters and fits quite well the data on the interfacial abruptness across double heterostructures in GaP/GaAs_x_P_1−x_/GaP nanowires. It can be used for any axial III–V nanowire heterostructures obtained through the vapor–liquid–solid method. It forms a basis for further developments in modeling the complex growth process and suppression of the interfacial broadening caused by the reservoir effect.

## 1. Introduction

III–V ternary nanowires (NWs) and heterostructures (HSs) of different types based on such NWs present an emerging class of nanomaterials that offer otherwise unattainable capabilities far beyond the 2D thin-film limit [1,2,3,4,5,6]. In particular, axial NW HSs BC/AC composed of lattice-mismatched III–V materials AC and BC can be grown without misfit dislocations if the NW radius is below the critical radius, which is typically in the order of a few tens of nanometers [7]. This property is due to the high surface-to-volume ratio of NWs allowing for the dislocation-free relaxation of elastic stress induced by lattice mismatch on the NW’s sidewalls. For the same reason, NWs containing axial HSs can be grown without dislocations on Si substrates, offering a promising way for the monolithic integration of III–V photonics with the Si electronic platform [2,3]. III–V NW HSs are ideal for the generation of and manipulation with quantum light [8,9,10]. Most III–V NWs are fabricated through different epitaxy techniques using the vapor–liquid–solid (VLS) method with a foreign metal catalyst (often Au [11]), which can be replaced with a group III metal (Ga or In) in the self-catalyzed approach [12]. A detailed review of modeling approaches for the VLS growth of III–V NWs can be found, for example, in Ref. [13].

Compositions of VLS III–V ternary NWs based on a group V intermix and NW HSs based on a group V interchange were studied experimentally in different material systems, including InAsSb [14,15,16,17,18], GaAsSb [18,19,20,21,22], InAsP [23,24,25,26,27,28,29] and GaAsP [30,31,32,33,34,35,36]. It is generally believed that the interfaces of axial NW HSs based on the group V interchange should be sharper compared to group-III-based NW HSs of a similar radius [37,38,39,40], because the reservoir effect caused by the accumulation of different elements in a catalyst droplet [13,18,20,27,33,35,36,37,38,39,40,41] is largely reduced for highly volatile group V atoms. However, the measured compositional profiles were almost atomically sharp only for VLS InAs/InP/InAs [27] and GaAs/GaAs_x_P_1−x_/GaAs [36] double NW HSs, while the interfaces of VLS GaP/GaAs/GaP [33] and GaP/GaAs_x_P_1−x_/GaP [35] double NW heterostructures were comparable or even wider than in GaAs/Al_x_Ga_1−x_As/GaAs NW HSs based on the group III interchange [39]. These examples showed that the compositional profiles across VLS axial NW HSs $A_{x_{0}}B_{1-x_{0}}C/A_{x_{1}}B_{1-x_{1}}C$ of a given radius are influenced by many factors, including the stationary compositions $x_{0}$ and $x_{1}$ and the crystallization rates of AC and BC pairs at the liquid–solid interface, rather than the total concentration of A and B atoms alone. The modeling strategies for the VLS growth of III–V ternary NWs and NW HSs are reviewed, for example, in Refs. [42] and [43]. In brief, they are based on the regular growth model [44], equilibrium [33,45] or nucleation-limited [46] and kinetic [47,48] models for the liquid–solid distribution x(y) connecting the composition of A_x_B_1−x_C NW with the relative fraction of atoms A in liquid y. The liquid composition y changes under time-dependent vapor fluxes of A and B atoms, producing a NW HS. With the known liquid–solid distribution x(y), and using some assumptions on the axial growth rate of a HS, one can compute the compositional profile x versus the vertical distance ξ [45,46]. According to the current view [49,50], the liquid–solid distribution of VLS ternary III–V NWs is close-to-equilibrium in the case of the group III intermix and kinetic in the case of the group V intermix, because the liquid–solid growth always occurs under group-III-rich conditions. This property was indirectly used in Ref. [33] for modeling the compositional profiles in self-catalyzed GaAs/AlGaAs NW HSs. However, it was not implied in the existing models for group-V-based NW HSs [18,27,44,46].

The liquid–solid distribution of VLS III–V ternary NWs based on the group V intermix is reduced to the one-parametric Langmuir–McLean shape [51] at high enough supersaturations in liquid. This purely kinetic distribution depends solely on the ratio of the effective liquid–solid incorporation rates of AC and BC pairs [49,50], denoted as $c_{l}$ in that which follows. The same Langmuir–McLean formula describes the equilibrium liquid–solid distribution for III–V ternaries with weak interactions between dissimilar pairs, such as AlGaAs, with a different parameter related to the ratio of the affinities of A and B atoms in liquid [39,45,46]. In Ref. [52], we presented the first attempt to develop a growth model for axial NW HSs based on the group V interchange, using a simplifying assumption on the time-independent total concentration of A and B atoms in liquid. Herein, we venture beyond this assumption and develop a self-consistent model for the compositional profiles across the axial NW HS $A_{x_{0}}B_{1-x_{0}}C/A_{x_{1}}B_{1-x_{1}}C$ with any stationary compositions $x_{0}$ and $x_{1}$. We use (i) quadratic dependences of the desorption rates of A_2_ and B_2_ dimers on the concentrations of A and B atoms in liquid [49,50,51,52,53] and (ii) linear dependences of the crystallization rates of AC and BC pairs at the liquid–solid interface on these concentrations [44]. The last assumption is a simplification of a complex growth process of a partial monolayer in VLS NWs [13], but should be valid for large enough droplets and high enough supersaturations in liquid. It yields the Langmuir–McLean shape of the liquid–solid distribution [44]. We explicitly calculate the time dependences of the concentrations of A and B atoms in liquid, the corresponding evolution of the droplet composition y and the NW composition x, the vertical coordinate ξ as a function of time and, finally, the compositional profile x(ξ). The model contains a minimum number of four parameters and can be used for the compositional modeling of any III–V NW HS grown using different epitaxy techniques. It fits well the compositional data on GaP/GaAs_x_P_1−x_/GaP NW HSs [35] and forms a basis for further advancements in the field.

## 2. Model

We considered the VLS growth of the axial NW HS $A_{x_{0}}B_{1-x_{0}}C/A_{x_{1}}B_{1-x_{1}}C$, where A and B atoms belong to group V and C atoms belong to group III, with the stationary compositions $x_{0}$ and $x_{1}$. In particular, $x_{0}=0$ corresponds to the VLS growth of the ternary NW section $A_{x_{1}}B_{1-x_{1}}C$ on the binary stem BC, and $x_{0}=0$ and $x_{1}=1$ correspond to the NW HS BC/AC composed of pure binary compounds. The total numbers of A and B atoms in a liquid droplet, $N_{A}$ and $N_{B}$, change in time according to:$$\frac{dN_{A}}{dt}=V_{A}-V_{A}^{des}-K_{A}\chi_{A},$$
(1)$$\frac{dN_{B}}{dt}=V_{B}-V_{B}^{des}-K_{B}\chi_{B}$$

Here, $V_{i}$ are the arrival rates into the droplet and $V_{i}^{des}$ are the desorption rates from the droplet of atoms i= A, B. The terms $K_{i}\chi_{i}$ are the linear sinks due to the crystallization of AC and BC pairs at the liquid–solid interface under the droplet, with $K_{i}$ as the corresponding crystallization or liquid–solid incorporation rates. The concentrations of i= A, B atoms in liquid $\chi_{i}$ were defined according to the following:(2)$$\chi_{A}=\frac{N_{A}}{N_{C}+N_{Au}+N_{A}+N_{B}},\;\;\;\chi_{B}=\frac{N_{B}}{N_{C}+N_{Au}+N_{A}+N_{B}},$$
where $N_{c}$ and $N_{Au}$ are the total numbers of group III atoms C and Au atoms in the droplet ($N_{Au}=0$ for self-catalyzed VLS growth). Linear crystallization rates used in Equation (1) neglected the fractions of A and B atoms rejected by the growing island or partial monolayer, and, hence, contained no interactions of AC and BC pairs in solid. This is valid only when the diffusion fluxes into the island are much larger than the rejected fluxes, which requires high enough supersaturations of liquid with respect to the ternary solid [44,47,48,49,50]. In this case, Equation (1) should be valid for both self-catalyzed and Au-catalyzed VLS growths, because the regular growth rate of the partial monolayer of a ternary NW is limited by the incorporation of group V atoms at $\chi_{i}\ll \chi_{C}$ [49,50]. Due to $N_{i}\ll N_{C}+N_{Au}$ for highly volatile group V atoms A and B, we could use the following:(3)$$\chi_{i}=\frac{N_{i}}{N_{tot}},\;\;i=A,B,\;\;\;N_{tot}≅N_{C}+N_{Au}=\frac{\pi R^{3}f(\beta )}{3\Omega_{L}}≅const.$$

The total number of atoms in the droplet $N_{tot}≅N_{C}+N_{Au}$ is related to the radius of the NW top R and the contact angle of the droplet β, with $\Omega_{L}$ as the elementary volume per atom in liquid and $f\left(\beta \right)$ as the known geometrical function of the droplet contact angle β [13].

The desorption rates $V_{i}^{des}$ of group V elements in the form of dimers A_2_ and B_2_ had to be proportional to $\chi_{i}^{2}$ and to the droplet surface area [13,49,50,52,53]. The surface diffusion of group V adatoms is usually negligible [53]. Therefore, we could write the following:(4)$$V_{i}=\frac{h}{\Omega_{S}}{\sigma_{i}v}_{i}\frac{2\pi R^{2}}{1+cos\beta },\;\;\;V_{i}^{des}=\frac{h}{\Omega_{S}}\sigma_{i}v_{i}^{des}\frac{2\pi R^{2}}{1+cos\beta }\chi_{i}^{2},\;\;i=A,B.$$

Here, h is the height of a NW monolayer, $\Omega_{S}$ is the elementary volume per III–V pair in solid, $v_{i}$ are the deposition rates of group V atoms ($v_{i}=2v_{i_{2}}$ for the deposition of dimers A_2_ and B_2_), $v_{i}^{des}$ are the known desorption factors ($v_{i}=2v_{i_{2}}^{dea}$) which can be found, for example, in Refs. [49,50,52,53], and $\sigma_{i}$ are the effective condensation coefficients or incorporation probabilities of group V species at the liquid surface [24,49,50,52]. The axial growth rate of a ternary NW, dξ/dt, where ξ is the vertical distance in monolayers, is related to the sum of the two crystallization rates according to the following:(5)$$\frac{\pi R^{2}h}{\Omega_{S}}\frac{d\xi }{dt}=K_{A}\chi_{A}+K_{B}\chi_{B}$$

Using Equations (3) to (5) in Equation (1), the governing equations describing the VLS growth and time-dependent composition of a ternary NW could be presented in the form shown below:(6)$$\frac{d\chi_{i}}{dt}=\gamma \left(\Phi_{i}-\Phi_{i}^{des}\chi_{i}^{2}-g_{i}\chi_{i}\right),\;\;\;i=A,B,$$
(7)$$\frac{d\xi }{dt}=g_{A}\chi_{A}+g_{B}\chi_{B}.$$

The parameters were given as follows:(8)$$\Phi_{i}=\frac{2\sigma_{i}}{1+cos\beta }v_{i},\;\;\Phi_{i}^{des}=\frac{2\sigma_{i}}{1+cos\beta }v_{i}^{des},\;g_{i}=\frac{\Omega_{S}}{\pi R^{2}h}K_{i},$$
and
(9)$$\gamma =\frac{3\Omega_{L}h}{\Omega_{S}Rf(\beta )}.$$

Clearly, $\Phi_{i}$ and $\Phi_{i}^{des}\chi_{i}^{2}$ stand for the direct impingement and desorption of group V species, respectively, while $g_{i}\chi_{i}$ describe the regular crystallization of AC and BC pairs producing a ternary solid. The sum of the two crystallization rates determines the vertical growth rate of a NW. With the known fluxes $\Phi_{i}$, our model contained only four control parameters, $\Phi_{A}^{des}$, $\Phi_{B}^{des}$, $g_{A}$ and $g_{B}$, describing the desorption rates of the A_2_ and B_2_ dimers and the crystallization rates of the AC and BC pairs at the growth interface, respectively, as shown in Figure 1. The desorption rates are the known functions of temperature and can be calculated for any III–V ternary [49,50,52,53]. However, the crystallization rates $g_{i}$ contain the generally unknown diffusion coefficients of A and B atoms in a quaternary liquid [49,50] and their values can be estimated only by fitting the experimental data on the interfacial profiles across NW HSs.

We noted that the regular growth rates $K_{i}$ were proportional to $\pi R^{2}$, hence, $g_{i}$ were independent of R [44]. The geometrical parameter γ was the same as in Refs. [27,39,44,45,46,52]. It decreased for larger R and β, showing that the strength of the reservoir effect increased for larger droplets. By solving Equation (6) after the group V flux commutation and using the boundary conditions before the commutation, we obtained the concentrations of A and B atoms $\chi_{A}$ and $\chi_{B}$ in liquid versus time. This gave the following liquid composition:(10)$$y=\frac{\chi_{A}}{\chi_{A}+\chi_{B}}$$
as a function of time. The solid composition in our model was obtained through the Langmuir–McLean formula:(11)$$x=\frac{g_{A}\chi_{A}}{g_{A}\chi_{A}+g_{B}\chi_{B}}=\frac{c_{l}y}{1+\left(c_{l}-1\right)y},\;c_{l}=\frac{g_{A}}{g_{B}}=\frac{K_{A}}{K_{B}},$$
and was obtained at the known y as a function of time. Finally, the interfacial profile ξ(t) was calculated through the integration of Equation (7) with the obtained time-dependences $\chi_{A}(t)$ and $\chi_{B}(t)$, which yielded the compositional profile x(ξ) versus the vertical coordinate.

Let us now discuss the novelty of the model with respect to the previous works. A similar growth model was earlier considered in Ref. [44]. However, it was not suited for ternaries based on the group V intermix because the desorption terms were assumed linear in $\chi_{i}$. The model of Ref. [27] entirely neglected the sinks of group V atoms through crystallization, assuming that the desorption fluxes of A and B atoms were much larger than the crystallization rates of AC and BC pairs, which is not true in the general case. The models of Refs. [39,45,46] and their generalizations [43] did not consider any desorption and, hence, were valid for HW HSs based on the group III interchange. The model of Ref. [52] used the assumption $g_{A}\chi_{A}+g_{B}\chi_{B}=\Phi_{A}-\Phi_{A}^{des}\chi_{A}^{2}+\Phi_{B}-\Phi_{B}^{des}\chi_{B}^{2}$; hence, $g_{A}\chi_{A}=x{(\Phi }_{A}-\Phi_{A}^{des}\chi_{A}^{2}+\Phi_{B}-\Phi_{B}^{des}\chi_{B}^{2})$ and $g_{B}\chi_{B}=(1-x){(\Phi }_{A}-\Phi_{A}^{des}\chi_{A}^{2}+\Phi_{B}-\Phi_{B}^{des}\chi_{B}^{2})$. This eliminated the crystallization rates $g_{A}\chi_{A}$ and $g_{B}\chi_{B}$ from the model and yielded $\chi_{A}+\chi_{B}=const$ at any time. Obviously, the condition of a time-independent total concentration of group V atoms in liquid could be broken in non-stationary regime of VLS growth under varying material fluxes. One could also use the approximation $g_{A}\chi_{A}+g_{B}\chi_{B}=\Phi_{C}$ in group-III-limited VLS growth of Au-catalyzed NWs [13,54] or under close-to-equilibrium conditions at a time-independent droplet volume [49,50]. The total flux of group III atoms $\Phi_{C}$ included their surface diffusion [13,49,50,54,55,56,57]. Our model was more general, because it considered the binary crystallization rates independently of the deposition and desorption fluxes and allowed for the accumulation or depletion of group V atoms in the droplet beyond the approximation of $\chi_{A}+\chi_{B}=const$.

## 3. Stationary State

Stationary solutions to Equation (6) were given as follows:

$$\chi_{i}^{s}=\frac{\sqrt{1+4\varphi_{i}\psi_{i}}-1}{2\psi_{i}},$$(12)$$\varphi_{i}=\frac{\Phi_{i}}{g_{i}},\;\psi_{i}=\frac{\Phi_{i}^{des}}{g_{i}}.$$
When $\varphi_{i}\psi_{i}=\Phi_{i}\Phi_{i}^{des}/g_{i}^{2}\ll 1$, we simply had $\chi_{i}^{s}≅\Phi_{i}/g_{i}$, that is, the stationary concentrations were proportional to the vapor fluxes of group V atoms. This corresponded to low desorption rates at a growth temperature or low deposition rates relative to the binary crystallization rates $g_{i}$. In the opposite case of $\Phi_{i}\Phi_{i}^{des}/g_{i}^{2}\gg 1$, corresponding to the desorption-limited VLS growth, the stationary concentrations $\chi_{i}^{s}≅\sqrt{\Phi_{i}/\Phi_{i}^{des}}$ were independent of $g_{i}.$ In our model, the low stability of group V atoms in the droplet ($\chi_{i}^{s}\ll 1$) could be due not only to the high desorption rates at elevated temperatures (large $\Phi_{i}^{des}\gg \Phi_{i}$), but also to high crystallization rates (large $g_{i}\gg \Phi_{i}$) even in the regimes with a low desorption.

The stationary solutions given by Equation (12) together with Equations (10) and (11) yielded the vapor–solid distribution $x_{s}(z)$ connecting the NW composition x with the fraction of A atoms in vapor z. Let us consider the VLS growth of a ternary NW from the vapor fluxes:(13)$$\Phi_{A}=\sigma_{A}\Phi_{tot}z,\;\Phi_{B}={\sigma_{B}\Phi }_{tot}(1-z),\;\Phi_{tot}=\frac{2}{1+cos\beta }v_{tot},$$
where $v_{tot}=v_{A}+v_{B}$ is the total deposition rate of group V atoms A and B. Using this in Equations (10) to (12), we obtained the following:(14)$$x_{s}=\frac{1}{1+G_{s}},\;\;G_{s}=u\frac{\sqrt{1+4\frac{\Gamma_{A}(1-z)}{c_{g}u}}-1}{\sqrt{1+4\Gamma_{A}z}-1},$$
with the below parameters:(15)$$\Gamma_{A}=\frac{{\sigma_{A}\Phi_{tot}\Phi }_{A}^{des}}{g_{A}^{2}},u=\frac{\Phi_{A}^{des}}{\Phi_{B}^{des}}\frac{1}{c_{l}^{2}},\;c_{g}=\frac{\sigma_{A}}{\sigma_{B}}.$$

Here, the parameter $\Gamma_{A}$ is similar to $\Phi_{A}\Phi_{A}^{des}/g_{A}^{2}$, and describes the fraction of desorbed atoms A. The parameter u accounts for the difference in the desorption rates of atoms A and B and the incorporation rates of AC and BC pairs at the liquid–solid interface. The parameter $c_{g}$ is related to the possible difference in the incorporation of atoms A and B at the droplet surface [24,30,49,50,52,55]. The vapor–solid distribution given by Equation (14) was simplified, because it neglected the interactions in a solid and contained no miscibility gaps for systems with strong interactions between dissimilar III–V pairs [50]. However, it described the transformation from the Langmuir–McLean kinetic shape (which became x=z at $c_{g}=1$) at low deposition rates of group V atoms or V/III ratios to a non-linear shape at high deposition rates or V/III ratios, caused by the enhanced desorption of the excessive group V species [14,49,50]. Indeed, at $\Gamma_{A}\to 0$, we had $G_{s}\to \left(1-z\right)/\left(c_{g}z\right)$, which gave the following Langmuir–McLean vapor–solid distribution:(16)$$x_{s}=\frac{c_{g}z}{1+\left(c_{g}-1\right)z}.$$

In the regime with high desorption of both group V elements at $\Gamma_{A}\to \infty$, $G_{s}\to {\left(u/c_{g}\right)}^{1/2}{\left[\left(1-z\right)/z\right]}^{1/2}$ and the vapor–solid distribution became the following:(17)$$x_{s}=\frac{\sqrt{z}}{\sqrt{z}+\upsilon \sqrt{1-z}},\nu =\sqrt{u/c_{g}}.$$

If atoms A incorporated into the liquid–solid interface much faster than atoms B, we could use u→∞ at $c_{l}^{2}\to 0$, in which case $G_{s}\to (2\Gamma_{A}/c_{g})\left(1-z\right)/(\sqrt{1+4\Gamma_{A}z}-1)$. In any case, the vapor–solid distribution became more non-linear in the regimes with an enhanced desorption of at least one group V element.

Figure 2 shows the measured vapor–solid distributions of GaAs_x_P_1−x_ NWs grown with Ga-catalyzed molecular beam epitaxy (MBE) at 630 °C under high V/III flux ratios > 16, as shown in Refs. [35] and [30]. The experimental curves were quite similar and demonstrated the difficult incorporation of As atoms into GaAsP in comparison to P atoms. Unfortunately, these limited datasets could be equally well-fitted using different expressions. The Langmuir–McLean fits required small values of $c_{g}=$ 0.247 and 0.335 to describe the lower As incorporation rate. The fits using Equation (14) required u→∞, suggesting that $c_{l}\ll 1$, which corresponded to a much faster transfer of P atoms from liquid to solid relative to As atoms. The large fitting values of $\Gamma_{A}=$ 20 and 9.5 suggested that most As atoms desorbed from the droplet. Therefore, it could not be stated which stationary state was closer to reality. More growth experiments with different III–V–V ternary NWs under different conditions (temperature, total V/III flux ratio, growth catalyst, etc.) are required for the accurate identification of the key factors influencing the shape of the vapor–solid distributions [49,50]. We noted, however, that the incomplete adsorption of As_2_ dimers on the liquid Ga surface in MBE was unlikely, and that the Langmuir–McLean curves with $c_{g}<1$ should not have corresponded to the negligible desorption of As and P species from liquid. The desorption of both As_2_ and P_2_ dimers from the droplet surface should have been significant at a high growth temperature of 630 °C [13,50,53]. Rather, the low-fitting values of $c_{g}$ in the Langmuir–McLean formula described in a qualitative way a larger fraction of desorbed As atoms in the Ga-catalyzed VLS growth of GaAsP NWs. This conclusion was earlier discussed in Ref. [50] in connection with the Langmuir–McLean fits of the stationary vapor–solid distributions of Ga-catalyzed GaAsP [30,32,35], Au-catalyzed GaAsP [58] and Au-catalyzed InAsP [24] VLS NWs grown at different temperatures through different epitaxy techniques. Furthermore, the Langmuir–McLean vapor–solid distributions were previously used for fitting the compositional data on GaAsP epilayers [59]. In GaAsP and InAsP material systems, the pseudo-binary interaction constants between IIIAs and IIIP pairs ω were noticeably lower than the critical value of 2 (in thermal units). For example, ω=0.641 for GaAsP at 630 °C [60]. Consequently, the exponential terms describing interactions in solid were weak and had little influence on the shape of the vapor–solid distributions [50]. This additional factor improved the correspondence of the Langmuir–McLean fits with the data on GaAsP and InAsP ternaries, even when the growth conditions were close-to-equilibrium [50].

## 4. Compositional Profiles

We now considered the time-dependent concentrations $\chi_{i}(t)$ under the varying vapor fluxes of A and B species. The most general case corresponded to a NW heterostructure $A_{x_{0}}B_{1-x_{0}}C/A_{x_{1}}B_{1-x_{1}}C$, with the two stationary compositions ${x_{s}=x}_{0}$ and $x_{1}$, stationary concentrations ${\chi_{i}^{s}=\chi }_{i}^{0}$ and $\chi_{i}^{1}$, obtained under the material fluxes $\Phi_{A}^{0}=\sigma_{A}\Phi_{tot}^{0}z_{0}$, $\Phi_{B}^{0}=\sigma_{B}\Phi_{tot}^{0}(1-z_{0})$, $\Phi_{A}^{1}=\sigma_{A}\Phi_{tot}^{1}z_{1}$ and $\Phi_{B}^{1}=\sigma_{B}\Phi_{tot}^{1}(1-z_{1})$. The stationary concentrations ${\chi_{i}^{s}=\chi }_{i}^{0}$ and $\chi_{i}^{1}$ were given using Equation (12) with these vapor fluxes. Exact solutions to Equation (6) with the initial conditions $\chi_{i}\left(t=t_{0}\right)=\chi_{i}^{0}$ were obtained with the below equations:(18)$$\chi_{i}\left(t\right)=\chi_{i}^{1}+\frac{2(\chi_{i}^{0}-\chi_{i}^{1})}{\left({1+\varepsilon }_{i}\right)exp\left(\frac{t-t_{0}}{\tau_{i}}\right)+1-\varepsilon_{i}},\;i=A,B,$$
with the following parameters:(19)$$\varepsilon_{i}=\frac{1+2\psi_{i}\chi_{i}^{0}}{1+2\psi_{i}\chi_{i}^{1}}=\sqrt{\frac{1+4\varphi_{i}^{0}\psi_{i}}{1+4\varphi_{i}^{1}\psi_{i}}},\;\;\;\frac{1}{\tau_{i}}=\gamma g_{i}\left(1+2\psi_{i}\chi_{i}^{1}\right).$$

The fraction of atoms A in liquid versus time was obtained from Equation (10). The solid composition versus time was obtained from Equation (11), which could equivalently be written as follows:(20)$$x(t)=\frac{1}{1+G(t)},\;G(t)=\frac{g_{B}\chi_{B}(t)}{g_{A}\chi_{A}(t)}=\frac{1}{c_{l}}\frac{\chi_{B}(t)}{\chi_{A}(t)}.$$

This gave the full picture of the VLS growth process and the droplet/NW composition as a function of time. Then, using Equation (18) in Equation (7) and integrating it with the initial condition $\xi \left(t=t_{0}\right)=\xi_{0}$, we obtained the vertical coordinate as a function of time:$$\xi \left(t\right)-\xi_{0}=F_{A}\left(t\right)+F_{B}(t),$$
(21)$$F_{i}\left(t\right)=g_{i}\chi_{i}^{1}\left(t-t_{0}\right)-\frac{g_{i}(\chi_{i}^{0}-\chi_{i}^{1})}{1-\varepsilon_{i}}\tau_{i}ln\left\{\frac{1+\varepsilon_{i}+\left(1-\varepsilon_{i}\right)exp\left[-\frac{\left(t-t_{0}\right)}{\tau_{i}}\right]}{2}\right\},\;i=A,B.$$

Equations (20) and (21) gave the compositional profile across the NW HS in the form of the implicit function, but not explicitly.

In the absence of the vapor flux of i= A or B atoms ($\Phi_{i}=\varphi_{i}=0$, $\chi_{i}^{1}=0$), Equation (18) was reduced to as shown:(22)$$\chi_{i}=\frac{\chi_{i}^{0}}{\left(1+\psi_{i}\chi_{i}^{0}\right)e^{\gamma g_{i}\left(t-t_{0}\right)}-\psi_{i}\chi_{i}^{0}}$$

This corresponded to the decrease in the concentration of i= A or B atoms from $\chi_{i}^{0}$ to zero after turning off their input. The depletion occurred due to the desorption of the residual atoms to vapor and their incorporation into solid. In the regime with negligible desorption ($\psi_{i}\to 0$), the droplet was depleted only due to the crystallization of AC or BC pairs:(23)$$\chi_{i}=\chi_{i}^{0}e^{-\gamma g_{i}\left(t-t_{0}\right)}.$$

In the absence of the desorption of the group V element i ($\psi_{i}\to 0$, $\varepsilon_{i}\to 1$), the function G(t) in Equation (20) was simplified to the below:(24)$$G=\frac{\Phi_{B}^{1}+(\Phi_{B}^{0}-\Phi_{B}^{1})e^{-\gamma g_{B}(t-t_{0})}}{\Phi_{A}^{1}+(\Phi_{A}^{0}-\Phi_{A}^{1})e^{-\gamma g_{A}(t-t_{0})}}.$$

The vertical coordinate ξ became the following:(25)$$\xi \left(t\right)=\xi_{0}+\left(\Phi_{A}^{1}+\Phi_{B}^{1}\right)\left(t-t_{0}\right)+\frac{\left(\Phi_{A}^{0}-\Phi_{A}^{1}\right)}{\gamma g_{A}}\left[1-e^{-\gamma g_{A}(t-t_{0})}\right]+\frac{\left(\Phi_{B}^{0}-\Phi_{B}^{1}\right)}{\gamma g_{B}}\left[1-e^{-\gamma g_{B}(t-t_{0})}\right].$$

These equations contained only the vapor fluxes $\Phi_{i}^{0}$ and $\Phi_{i}^{1}$ before and after the group V flux commutation. The concentrations of atoms A and B in liquid were given using $\chi_{i}^{0}=\Phi_{i}^{0}/g_{©}$ and $\chi_{i}^{1}=\Phi_{i}^{1}/g_{i}$.

Figure 3 show the solutions obtained from Equations (20), (24) and (25) without the desorption of both group V elements from the droplet. The coefficient in the Langmuir–McLean vapor–solid distribution $c_{g}$ was set to 0.247, as in Figure 2, for the data of Ref. [35]. The parameter γ was set to 0.000758, which corresponded, for example, to a 110 nm radius GaAsP NW with β= 120° [52]. We used the total fluxes such that $2/(1+cos\beta )]\sigma_{B}v_{tot}^{0}=[2/(1+cos\beta )]\sigma_{B}v_{tot}^{1}=$ 1 ML/s. At $z_{0}=0$, $z_{1}=0.5$, the input fluxes were given as follows: $\Phi_{A}^{0}=0$, $\Phi_{B}^{0}=$ 1 ML/s and $\Phi_{A}^{1}=$ 0.1235 ML/s, $\Phi_{B}^{1}=$ 0.5 ML/s. At $z_{0}=0.5$ and $z_{1}=1$. The input fluxes were given as follows: $\Phi_{A}^{0}=$ 0.1235 ML/s, $\Phi_{B}^{0}=$ 0.5 ML/s and $\Phi_{A}^{1}=$ 0.247 ML/s and $\Phi_{B}^{1}=$ 0. We chose $g_{B}=$ 100 ML/s, corresponding to $\chi_{B}^{0}=\Phi_{B}^{0}/g_{B}=$ 0.01 at $z_{0}=0$. Three different $c_{l}=$ 0.1, 1 and 10 then yielded $g_{A}=$ 10, 100 and 1000 ML/s and $\chi_{A}^{1}=\Phi_{A}^{1}/g_{A}=$ 0.0247, 0.00247 and 0.000247, respectively, at $z_{1}=0.5$. Figure 3a,b show that both x and y became more abrupt functions of time for larger $c_{l}$. The time-dependent profiles were largely influenced by the initial NW composition $x_{0}$, which equaled zero at $z_{0}=0$ and 0.2 at $z_{0}=0.5$. At a small $c_{l}$ of 0.1, the x(t) dependence in the direct transition was much sharper if we started from a non-zero $x_{0}$. For example, As atoms were more stable in liquid compared to P atoms [27,52], corresponding to $c_{l}<1$, or even $c_{l}\ll 1$, for GaAs_x_P_1−x_ NWs. This explained why the GaAs_x_P_1−x_/GaAs/GaAs_x_P_1−x_ interfaces of Ref. [36] were much sharper than the GaP/GaAs_x_P_1−x_/GaP interfaces of Ref. [35], even if they were grown in the self-catalyzed VLS mode under very similar MBE conditions and temperatures (630 °C in Ref. [35] and 640 °C in Ref. [36]). This important observation was recognized in Ref. [52] using a different model. The effect remained if the desorption of As and P atoms was included (see Figure 3 below). According to Figure 3c, the vertical distance versus time was composed of two almost linear segments, with the abrupt change in the slope angle at the flux commutation. This corresponded to higher growth rates of NW sections with smaller fractions x of AC material in the ternary A_x_B_1−x_C NW. Smaller x were obtained at lower z, which yielded higher total fluxes of group V atoms due to $c_{g}<1$ with our parameters. The growth rates of NW HSs at $z_{0}=0$, $z_{1}=0.5$ were, therefore, much higher than at $z_{0}=0.5$, $z_{1}=1$. This property strongly affected the compositional profiles x(ξ) in Figure 3d, which became significantly thinner for lower growth rates at larger z. Otherwise, the shapes of the compositional profiles x(ξ) in Figure 3d were quite similar to the x(t) dependences in Figure 3a. In particular, we obtained broader heterointerfaces at a small $c_{l}$ of 0.1 and $z_{0}=0$, that is, when the NW HSs started from a pure BC segment. At $z_{0}=0.5$, the interfacial abruptness at $c_{l}=0.1$ largely improved, particularly for the direct transition. The interfaces at $c_{l}=1$ and 10 became almost indistinguishable.

Figure 4 shows the same dependences obtained from Equations (18) to (21), where the desorption of A and B species was included. In this case, we used $c_{g}=1$, corresponding to the 100% incorporation of both A_2_ and B_2_ dimers from vapor into the droplet. The parameter γ equaled 0.000758, as in Figure 3. The parameter $c_{l}$ equaled 0.1, meaning that atoms A were more stable in the droplet than atoms B as for As and P atoms in the Ga-catalyzed growth of GaAs_x_P_1−x_ NWs [35,36]). We assumed that $\Phi_{A}^{des}=\Phi_{B}^{des}$. The total input fluxes of group V atoms were fixed at $\sigma_{A}\Phi_{tot}^{0}=\sigma_{B}\Phi_{tot}^{0}=\sigma_{A}\Phi_{tot}^{1}=\sigma_{B}\Phi_{tot}^{1}=$ 1 ML/s. At $z_{0}=0$, $z_{1}=0.5$, the vapor fluxes of A and B atoms were given as follows: $\Phi_{A}^{0}=0$, $\Phi_{B}^{0}=$ 1 ML/s and $\Phi_{A}^{1}=$ 0.5 ML/s, $\Phi_{B}^{1}=$ 0.5 ML/s. At $z_{0}=0.5$, $z_{1}=1$, the fluxes were given as follows: $\Phi_{A}^{0}=$ 0.5 ML/s, $\Phi_{B}^{0}=$ 0.5 ML/s and $\Phi_{A}^{1}=$ 1 ML/s, $\Phi_{B}^{1}=$ 0. We chose $g_{B}=$ 100 ML/s and, hence, $g_{A}=c_{l}g_{B}=$ 10 ML/s. Three different $\Gamma_{A}=$ 1, 5 and 20 yielded $\Phi_{A}^{des}=\Phi_{B}^{des}=$ 100, 500 and 2000 ML/s, respectively. The stationary concentrations of group V atoms in liquid at different transitions are given in Table 1. A larger $\Gamma_{A}$ corresponded to higher desorption rates of both A_2_ and B_2_ dimers from the droplet, which was why their concentrations decreased with $\Gamma_{A}$. The concentrations of B atoms were much lower than for A atoms in all cases due to a small $c_{l}$ of 0.1. At the same desorption coefficients ($\Phi_{A}^{des}=\Phi_{B}^{des}$), lower concentrations of B atoms were explained solely by their faster transfer from liquid to solid. The general trend of sharpening the NW HSs by using the ternary NW stem $A_{x_{0}}B_{1-x_{0}}C$ with $x_{0}=0.2-0.45$ for growing a pure AC segment on its top remained the same as in Figure 3.

The NW HSs in Figure 4 became thinner for higher desorption rates. Enhanced desorption improved the interfacial abruptness, as clearly seen from Figure 4d. This important conclusion could be understood as follows: The reservoir effect, caused by the accumulation of, for example, atoms B, in a catalyst droplet when growing a pure AC NW segment [13,18,20,27,33,35,36,37,38,39,40,41,52] contributed to the interfacial broadening only through the crystallization of a BC pair together with AC pairs at the liquid–solid interface. In the VLS growth regimes with low desorption, almost all atoms B would be transferred from liquid to solid, while most of these atoms would evaporate without influencing the interfacial abruptness when their desorption was enhanced. Similar considerations were used in Ref. [33] for sharpening the self-catalyzed GaAs/GaP NW HSs by switching all material fluxes for a short time at the group V flux commutation.

To demonstrate the effect more clearly, we noted that the $\xi \left(t\right)$ curves in Figure 3c and Figure 4c were almost linear, with the axial growth rate changing very abruptly upon the flux commutation. In this case, we could use the approximation of a time-independent axial growth rate of a NW HS at a given transition:(26)$$v=g_{A}\chi_{A}+g_{B}\chi_{B}≅const$$

Equations (11) and (7) were then reduced to $x≅g_{A}\chi_{A}/v$ and $\xi -\xi_{0}≅v(t-t_{0})$. Using this in Equation (18), we obtained the simplified expression for the compositional profile:$$x≅x_{1}+\frac{{2(x}_{0}-x_{1})}{\left(1+\varepsilon \right)exp\left(\frac{\xi -\xi_{0}}{\Delta \xi }\right)+1-\varepsilon },$$
(27)$$\varepsilon =\frac{1+2\omega x_{0}}{1+2\omega x_{1}},\;\;\Delta \xi =\frac{v}{\gamma g_{A}(1+2\omega x_{1})},\;\;\omega =\frac{\psi_{A}v}{g_{A}}=\frac{\Phi_{A}^{des}v}{g_{A}^{2}}.$$

This formula contained the experimentally measurable stationary compositions $x_{0}$ and $x_{1}$; the axial growth rates v could also be measured for each transition [27,33,35,36]. The parameter γ given by Equation (9) was known from the NW geometry (R and β). The crystallization rate of AC pairs $g_{A}$ and, hence, ω, were generally unknown, because $g_{A}$ contained the unknown diffusion coefficient of atoms A in Ga or Ga–Au liquid [50]. However, the characteristic interfacial abruptness Δξ could be estimated from fitting the experimental compositional profiles for a given material system. From Equation (27), the interfacial abruptness at a given $v/\gamma g_{A}$ improved for larger ω, and ω increased for a higher desorption rate $\Phi_{A}^{des}$. Figure 5 shows that the double NW HS with $x_{0}=0$ and $x_{1}=0.5$ became sharper when desorption was enhanced, particularly at the direct transition.

## 5. Theory and Experiment

Figure 6a shows the measured compositional profiles in double NW HSs ${GaP/GaAs}_{x_{1}}P_{1-x_{1}}/GaP$ with different stationary compositions $x_{1}$ from 0.093 to 0.65, as shown in Figure 1 (Data 1 from Ref. [35]). The NW HSs were grown using Ga-catalyzed MBE at 630 °C on Si(111) substrates. The Ga deposition rate was fixed in all experiments. GaP NW stems as well as pure GaP sections in the NW HSs were grown at a V/III flux ratio of eight, which gave an average axial growth rate of 1.362 ML/s. Different fractions of GaAs in ternary GaAsP sections were achieved by changing the As/P flux ratio from 0.5 to 10. The total V/III flux ratio was kept at 16 for As/P ratios from 0.5 to 2, and at 32 for As/P ratios larger than 2. The average radius of the NW HSs was 110 nm, and the contact angle was estimated at 120° in Ref. [52], yielding γ= 0.000758 from Equation (9). The dashed lines in Figure 6a show the best fits obtained from our model without desorption. These curves were calculated using Equations (20), (24) and (25), where we took into account the increase in the total flux of group V atoms from 1.362 ML/s for pure GaP to 2.724 for ${GaAs}_{x_{0}}P_{1-x_{0}}$ with low $x_{1}\leq 0.367$ at As/P ratios ≤2 and to 5.448 for higher $x_{1}$ at larger As/P ratios. The parameter $c_{g}$ was fixed at 0.247, according to the best fit obtained through the Langmuir–McLean vapor–solid distribution in Figure 1. Other parameters are summarized in Table 2. Figure 6b shows the vertical distance ξ versus time, obtained with these parameters and used in calculations of the compositional profiles. We noted that the growth time of the GaAsP segments was 36 s and 18 s for As/P ratios ≤2 and >2, respectively [35]. The model without desorption required a low $c_{g}=0.247$ to fit the stationary vapor–solid distributions in Figure 2. Therefore, the increase in the As flux with respect to the P flux did not produce the required increase in the axial growth rate, according to Figure 6b.

As mentioned above, the desorption of As_2_ and P_2_ dimers from the Ga droplet should have been significant at 630 °C. The solid lines in Figure 6a show the best fits obtained from Equations (27) with desorption at $c_{g}=$ 1 and the same γ of 0.000758. In these fits, the axial growth rate v was calculated as the linear interpolation of the HS height with the known growth times (36 and 18 s). The fitting parameters are summarized in Table 3. The use of the general model given by Equations (18) to (21) resulted in almost identical curves. It could be seen that the calculated compositional profiles were narrower in the model with desorption. Figure 7 shows the comparison of the fits to the data of Ref. [35] obtained in Ref. [52] and using our model with desorption. Additionally, we showed the fits to the data of Ref. [36], where pure GaAs segments were grown on ternary GaAs_0.6_P_0.4_ stems through Ga-catalyzed MBE at 640 °C under conditions similar to Ref. [35]. Pure GaAs segments were achieved through the termination of the P flux ($x_{1}=1$). These NWs had a uniform radius of 30 nm, and the contact angle estimated in Ref. [52] was approximately 135°. The corresponding value of γ equaled 0.00135, according to Equation (9). The Ga droplets on top of these NWs were smaller than in Ref. [35]. Consequently, the characteristic interfacial abruptness $\Delta \xi_{c}=v/\gamma g_{A}$ decreased from ~40 MLs (see the fitting parameters for the data of Ref. [35] in Table 3) to 25 MLs for Ref. [36]. The fitting value of ω=2 was the same for the data of Refs. [35] and [36]. This seemed plausible, because both GaAsP NW HSs were grown using Ga-catalyzed MBE at similar temperatures of 630–640 °C.

According to Figure 6 and Figure 7, all models considered provided reasonable fits to the data. Figure 6 clearly demonstrates the narrowing effect in GaAsP NW HSs, which started from ternary GaAs_0.6_P_0.4_ stems, as discussed above (see Figure 2) and in Ref. [52]. The fits of the sharp NW HSs GaAs_0.6_P_0.4_/GaAs/GaAs_0.6_P_0.4_ of Ref. [36] in Figure 7 using the model of Ref. [52] and using the model of this work were almost indistinguishable. The reverse interface in the transition from pure GaAs to GaAs_0.6_P_0.4_ was broader than in the direct transition. This was explained by the higher stability of As atoms in liquid Ga relative to P atoms, which was why the removal of As atoms took more time [36,52]. Considering the fits to the compositional profiles of GaP/GaAs_x_P_1−x_/GaP NW HSs in Figure 6a, we believe that the curves with the desorption of both group V atoms at $c_{g}=$ 1 were closer to reality and were obtained for the experimental axial growth rates at each transition. The fits without desorption at $c_{g}=$ 0. 247 underestimated the axial growth rates for large As fractions in vapor, as discussed above. From Figure 7, the compositional profiles at the direct transition from GaP to GaAs_x_P_1−x_ were better fitted with the model of Ref. [52], while the reverse interfaces were better fitted with the model of this work. The model of Ref. [52] obviously overestimated the interfacial widths for all reverse transitions. In Ref. [52], we also used the Langmuir–McLean liquid–solid distribution, and obtained the fits shown in Figure 6 at $c_{l}=0.1$. According to Table 2, the best fits to the same data using the model without desorption were obtained at $g_{P}=$ 200 ML/s and $g_{As}=$ 50–65 ML/s, yielding $c_{l}=g_{As}/g_{P}$ from 0.25 to 0.325. In the model with desorption, from ${\Delta \xi }_{c}=v/\gamma g_{A}≅$ 40 MLs and γ= 0.000758, we obtained $g_{A}≅41$ and, hence, $c_{l}≅$ 0.2. Therefore, all models predicted a faster transfer of P atoms from Ga liquid to solid relative to As atoms, with $c_{l}$ ranging from 0.1 to ~0.3.

## 6. Conclusions

In conclusion, a self-consistent model for compositional profiles across VLS III–V NW HSs based on the group V interchange was developed. The model essentially relied on only one assumption on the linear dependence of the crystallization rates of AC and BC pairs on the concentrations $\chi_{A}$ and $\chi_{B}$, which automatically led to the Langmuir–McLean liquid–solid distribution. The quadratic dependence of the desorption rates of A and B atoms on their concentrations should have held regardless of the growth catalyst [13]. No further simplifications, such as $\chi_{A}+\chi_{B}=const$ in Ref. [52], were used in our treatment. The model allowed for the accumulation or depletion of group V atoms in liquid and the treatment of different non-stationary processes such as emptying the droplet upon the growth interruption. Simple analytical solutions obtained for the time-dependent concentrations of group V elements in the droplet, liquid and solid composition and the height of a HS provided the full description of the growth process. However, the coordinate-dependent compositional profile was obtained only implicitly. The model fit quite well the data on axial NW HSs in the GaAsP system and could be extended to other III–V ternary materials. The unknown crystallization rates of AC and BC pairs at the growth interface could be deduced from fitting the data on the interfacial profiles in III–V NW HSs.

We revealed some general properties of VLS NW HSs based on the group V interchange and their interfacial abruptness. In particular, the interfaces of VLS A_x_B_1−x_C NW HSs with a more stable group V element A greatly improved if started from ternary A_x_B_1−x_C and transitioned to pure AC rather than starting from pure BC and transitioning it to ternary A_x_B_1−x_C. Higher desorption rates of both group V elements suppressed the interfacial broadening due to the reservoir effect because the desorbed species did not contribute into crystallization at the growth interface. These factors should have been carefully considered in the growth experiments and could be used for obtaining a more abrupt interface in VLS NW HSs. The presented approach required refinements in several respects, such as including the rejected species and, hence, pseudo-binary interactions in solid and miscibility gaps [42,43,50], as well as treating the NW monolayer progression with the depletion of group V atoms and a possible “stopping size”, where the supersaturation of liquid drops to zero [13,61]. This process was extremely complex even for binary VLS III–V NWs. We plan to consider these interesting effects and their impact on the compositional profiles in III–V NW HSs in forthcoming work.

## Figures and Tables

**Figure 1 nanomaterials-14-00821-f001:**
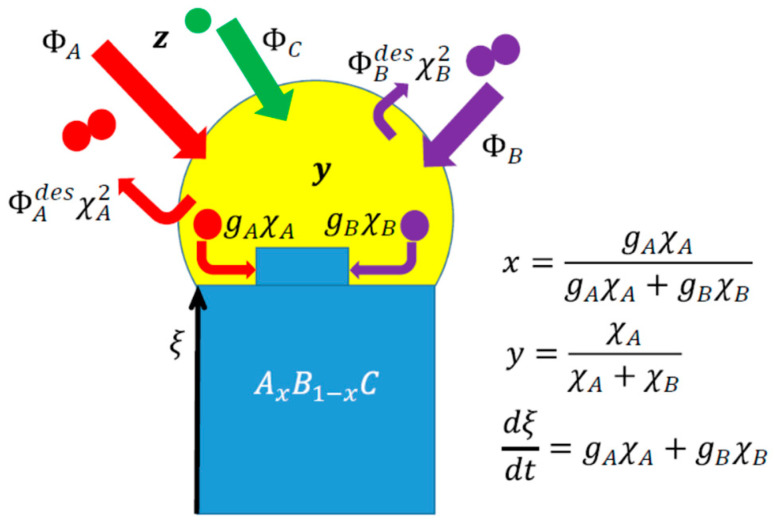
Illustration of the model parameters for VLS growth of III–V ternary NW HS based on group V intermix from vapor fluxes $\Phi_{A}$, $\Phi_{B}$ and $\Phi_{c}$. The desorption rates of group V dimers $\Phi_{i}^{des}\chi_{i}^{2}$ are proportional to the squared concentrations of group V atoms in liquid. The crystallization rates of dissimilar III–V pairs $g_{i}\chi_{i}$ scale linearly with $\chi_{i}$. The vapor composition z is determined by the fluxes. The solid composition x is given by the ratio of the crystallization rate of AC pairs over the total crystallization rate, which equals the axial NW growth rate dξ/dt.

**Figure 2 nanomaterials-14-00821-f002:**
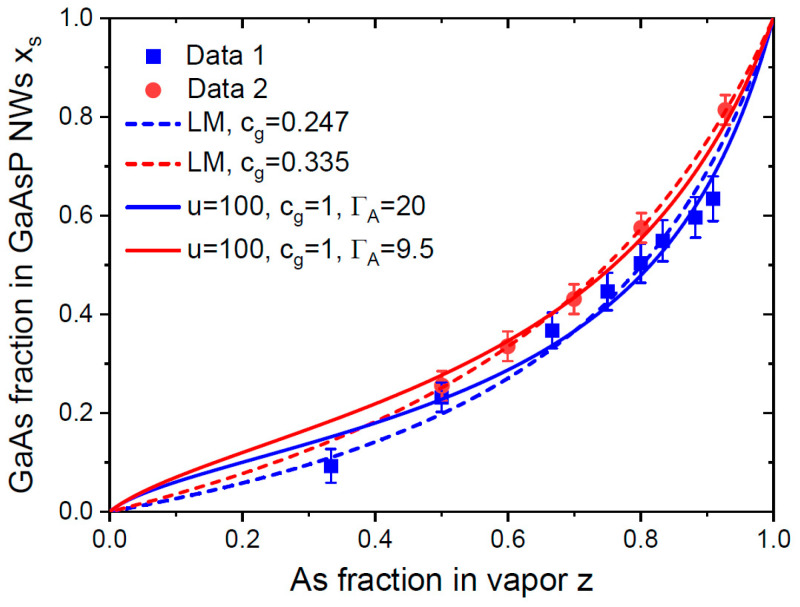
Stationary vapor–solid distributions of GaAsP NWs grown using Ga-catalyzed MBE at 630 °C in Ref. [35] (Data 1) and [30] (Data 2) (symbols), fitted using the Langmuir–McLean (LM) formula given through Equation (16) with $c_{g}$ given in the legend (dashed lines) and Equation (14) at $c_{g}=1$ and other parameters given in the legend (solid lines).

**Figure 3 nanomaterials-14-00821-f003:**
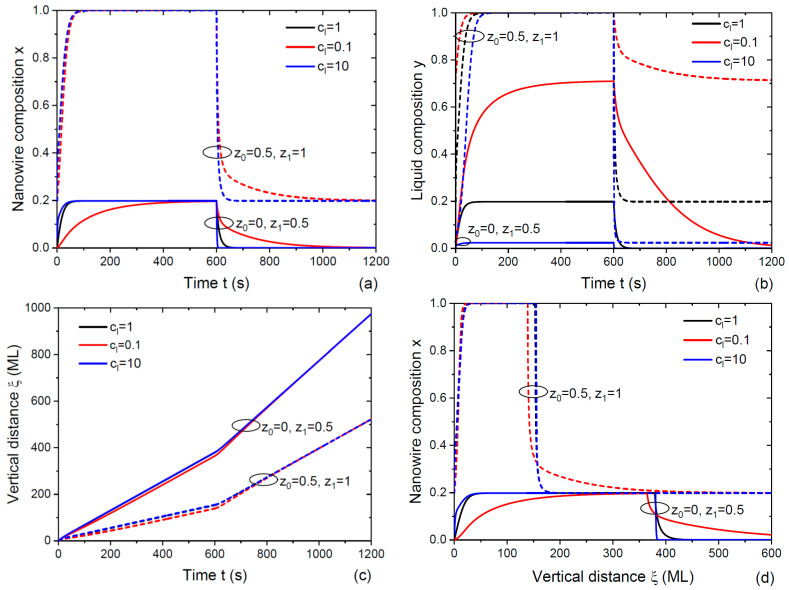
(**a**) Time-dependent NW composition, (**b**) liquid composition, (**c**) vertical distance and (**d**) coordinate-dependent compositional profiles across double NW HSs with different stationary fractions of group V atoms A in vapor: $z_{0}=0$, $z_{1}=0.5$ (solid lines) and $z_{0}=0.5$, $z_{1}=1$ (dashed lines). The curves were obtained from Equations (20), (24) and (25) without desorption at $c_{g}=$ 0.247, γ= 0.000758 and three different $c_{l}=$ 1, 0.1 and 10. Other parameters are given in the main text.

**Figure 4 nanomaterials-14-00821-f004:**
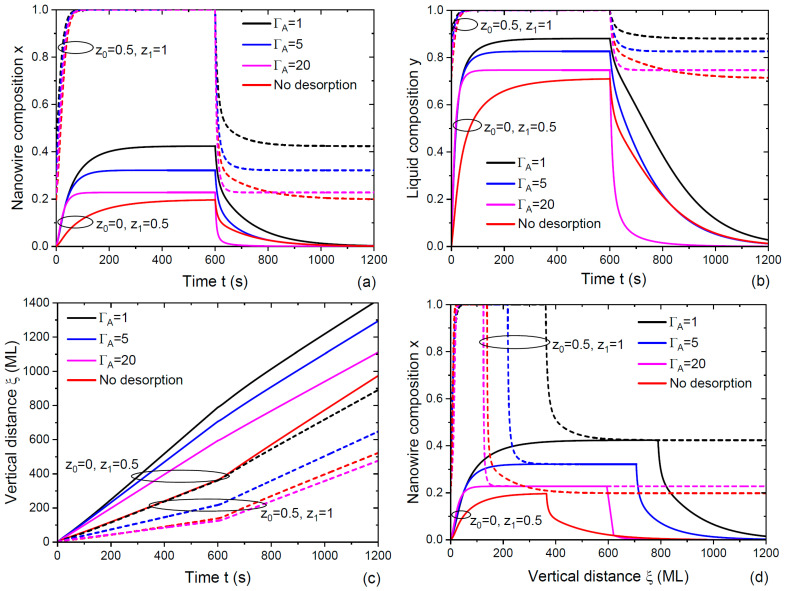
Same as Figure 3 with the desorption of A_2_ and B_2_ dimers from the droplet: (**a**) Time-dependent NW composition, (**b**) liquid composition, (**c**) vertical distance and (**d**) coordinate-dependent compositional profiles across double NW HSs. The curves were obtained from Equations (18) to (21) at $c_{g}=$ 1 (equivalent incorporation of A and B species into the droplet), $c_{l}=$ 0.1 (high stability of A atoms in liquid relative to B atoms), γ= 0.000758 and three different $\Gamma_{A}=$ 1, 5 and 20. Larger $\Gamma_{A}$ corresponded to higher desorption rates of both group V elements. Other parameters are given in the main text and in Table 1. The curves without desorption at $c_{g}=$ 0.247 and $c_{l}=$ 0.1 from Figure 3 were included for comparison.

**Figure 5 nanomaterials-14-00821-f005:**
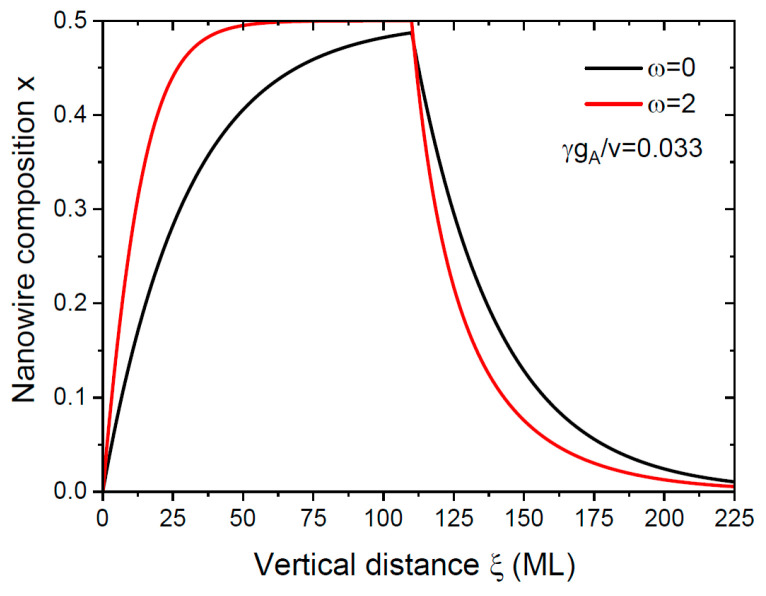
Compositional profiles across double NW HS BC/A_0.5_B_0.5_C/BC without (ω=0) and with (ω=2) desorption of group V elements, obtained from Equation (27) at a fixed $\gamma g_{A}/v$ of 0.033, corresponding to the same axial growth rate at both transitions.

**Figure 6 nanomaterials-14-00821-f006:**
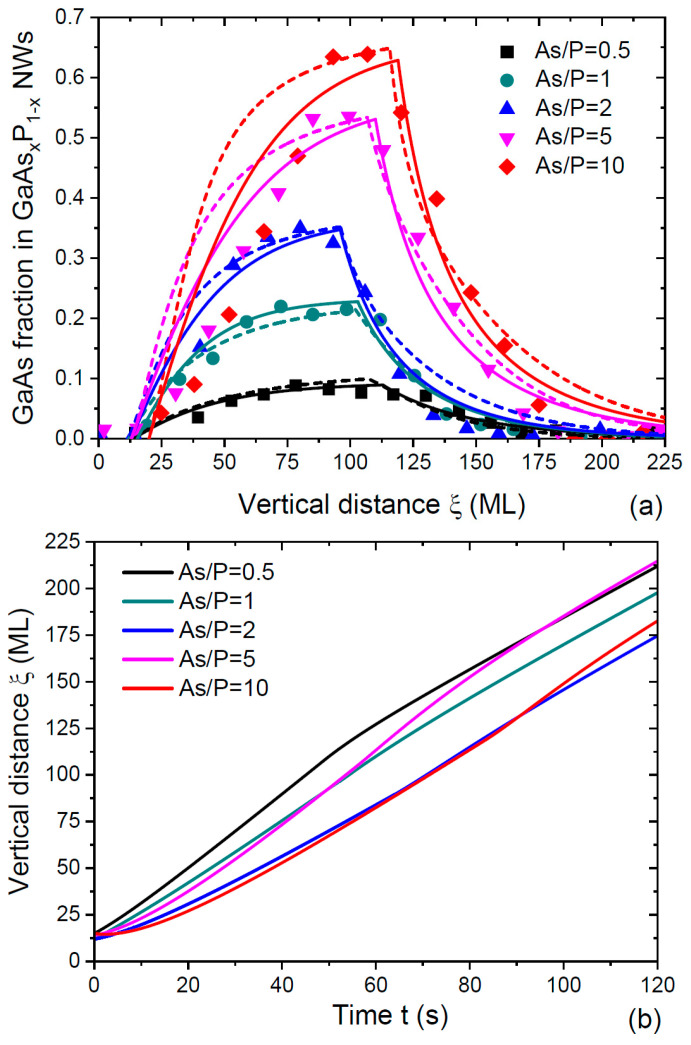
(**a**) Compositional profiles across double NW HSs ${GaP/GaAs}_{x_{1}}P_{1-x_{1}}/GaP$ with different stationary compositions $x_{1}$, grown using Ga-catalyzed MBE at 630 °C under different As/P ratios, as shown in the legend [35] (symbols). Dashed lines show the fits obtained from the model without desorption at $c_{g}=$ 0.247 and γ= 0.000758, with other parameters summarized in Table 2. Solid lines show the fits obtained from the model with desorption at $c_{g}=$ 1 and γ= 0.000758, with other parameters summarized in Table 3. (**b**) Vertical distance versus time at different As/P ratios used in calculations of the compositional profiles without desorption in (**a**).

**Figure 7 nanomaterials-14-00821-f007:**
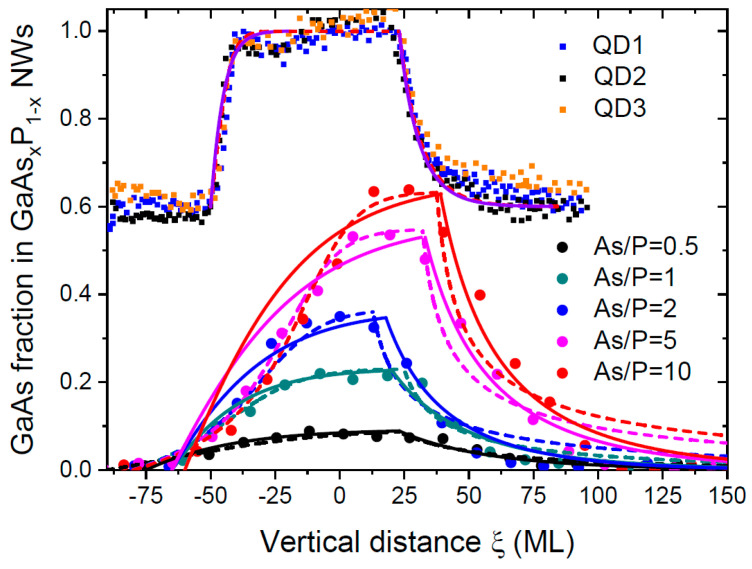
Same data as in Figure 6a for double NW HSs GaP/GaAsP/GaP with different compositions of GaAs obtained under different As/P ratios, as shown in the legend [35], along with much sharper NW HSs GaAs_0.6_P_0.4_/GaAs/GaAs_0.6_P_0.4_ for three GaAs quantum dots QD1, Q2 and QD3, obtained in Ref. [36] using very similar Ga-catalyzed MBE process (symbols). Dashed lines show the fits of Ref. [52]. Solid lines are the fits obtained within our model with desorption. The fits using the model of Ref. [52] and using the model of this work for the sharp NW HSs of Ref. [36] were almost indistinguishable.

**Table 1 nanomaterials-14-00821-t001:** Stationary concentrations of A and B atoms for NW HSs shown in Figure 4.

	$\Gamma_{A}$	1	5	20
$z_{0}=0$ $z_{1}=0.5$	$\chi_{A}^{0}$	0	0	0
	$\chi_{A}^{1}$	0.0366	0.0232	0.0135
	$\chi_{B}^{0}$	0.0099	0.00954	0.00854
	$\chi_{B}^{1}$	0.00498	0.00488	0.00458
$z_{0}=0$.5 $z_{1}=1$	$\chi_{A}^{0}$	0.0366	0.0232	0.0135
	$\chi_{A}^{1}$	0.0618	0.0358	0.020
	$\chi_{B}^{0}$	0.00498	0.00488	0.00458
	$\chi_{B}^{1}$	0	0	0

**Table 2 nanomaterials-14-00821-t002:** Fitting parameters for GaP/GaAs_x_P_1−x_/GaP NW HSs in Figure 6 without desorption.

Direct Transition from GaP to GaAs_x_P_1−x_					
As/P flux ratio	0.5	1	2	5	10
z	0.333	0.5	0.667	0.833	0.909
$\Phi_{tot}^{1}$ (ML/s)	2.724	2.724	2.724	5.448	5.448
$\Phi_{P}^{0}$ (ML/s)	1.362	1.362	1.362	1.362	1.362
$\Phi_{As}^{0}$ (ML/s)	0	0	0	0	0
$\Phi_{As}^{1}+\Phi_{P}^{1}$ (ML/s)	2.04	1.762	1.447	2.11	1.596
$g_{As}$ (ML/s)	65	65	55	60	50
$\chi_{As}^{1}$	0.0034	0.00662	0.0098	0.020	0.022
$g_{P}$ (ML/s)	200	200	200	200	200
$\chi_{P}^{1}$	0.0091	0.0068	0.0045	0.0045	0.0025
Reverse transition from GaAs_x_P_1−x_ to GaP					
$\Phi_{P}^{0}$ (ML/s)	1.817	1.362	0.907	0.91	0.496
$\Phi_{P}^{1}$ (ML/s)	1.362	1.362	1.362	1.362	1.362
$\Phi_{As}^{0}$ (ML/s)	0.224	0.336	0.448	1.12	1.224
$\Phi_{As}^{1}$ (ML/s)	0	0	0	0	0

**Table 3 nanomaterials-14-00821-t003:** Fitting parameters for GaP/GaAs_x_P_1−x_/GaP NW HSs in Figure 6 with desorption.

Direct Transition from Pure GaP to GaAs_x_P_1−x_					
As/P flux ratio	0.5	1	2	5	10
v (ML/s)	2.5	2.5	2.5	5	5
$x_{0}$	0	0	0	0	0
$x_{1}$	0.0927	0.232	0.367	0.549	0.65
Reverse transition from GaAs_x_P_1−x_ to pure GaP					
v (ML/s)	1.25	1.25	1.25	1.25	1.25
${\Delta \xi }_{c}=v/\gamma g_{As}$ (MLs)	40	40	35	45	45
ω	2	2	2	2	2

## Data Availability

Data are contained within the article.

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
