# Peer review of "Self-Consistent Model for the Compositional Profiles in Vapor–Liquid–Solid III–V Nanowire Heterostructures Based on Group V Interchange"

_nanomaterials, 2024, doi:10.3390/nano14100821_

Round 1
Reviewer 1 Report
Comments and Suggestions for Authors
The manuscript by Dubrovskii et al. presents a detailed study on the simulation of compositional profiles in VLS III-V nanowire heterostructures based on group V interchange through self-consistent modelling.
The topic has been largely discussed in literature both from an experimental and a theoretical standpoint. However, the author proposes here a refined modelling framework which is better suited for a more general description of the system. Even though at the moment the comparison with the experiment does not revel striking advances with respect to published results.
I believe that the paper is scientifically sound and the results of interest, even though mostclikely for a specialized audience. Overall, I think the paper should be published since the developed model is very clearly presented and discussed and it could be useful for people working in the field.
Reviewer 2 Report
Comments and Suggestions for Authors
Vladimir G. Dubrovskii reports an article paper entitled “Self-consistent model for the compositional profiles in VLS III-V nanowire heterostructures based on group V interchange”. The title attracts the readers but the contents seem to be different from the title and are not friendly to the readers.
First, it is hard for readers to understand the model. The author should present supporting schematic figures with parameters in the formulas to support the readers to understand the model. Second, the author claims that the paper presents a fully self-consistent model. However, the analysis used parameters obtained from the data fitting. It seems not to be “a fully self-consistent”. The author should clearly present the boundary condition, used parameters from the data, and what information can be obtained from this model. In addition, what is the advantage of the model analysis and what is the new information obtained by analysis using this model? The authors should clearly show the contribution of the paper to the readers.
For the above-mentioned reasons, reconsideration of the title and adding more valuable information about the model are recommended.
Comments on the Quality of English LanguageEnglish should be improved for the readers to easily understand the contents.
Reviewer 3 Report
Comments and Suggestions for Authors
The manuscript, titled "Self-consistent Model for Compositional Profiles in VLS III-V Nanowire Heterostructures Based on Group V Interchange," presents a self-consistent model for calculating compositional profiles during vapor-liquid-solid growth of axial nanowire heterostructures. The model's validation with data on interfacial abruptness in GaP/GaAsxP1-x/GaP nanowires demonstrates its effectiveness. The authors assert its applicability to various axial III-V nanowire heterostructures, paving the way for enhanced growth process modeling and interfacial broadening suppression. While the paper is well-written with justified discussion and conclusions, one concern arises from its comparison with another model by the same authors, Ref. [52], which is also under review. This comparison complicates assessing whether the paper introduces a new scientific model or merely implements an existing one.
Round 2
Reviewer 2 Report
Comments and Suggestions for Authors
Vladimir G. Dubrovskii reports an article paper entitled “Self-consistent model for the compositional profiles in VLS III-V nanowire heterostructures based on group V interchange”. Although I feel there are still some issues to be improved for friendly to the readers, the manuscript has been revised according to the reviewers’ suggestion. I think the manuscript would be reviewed and discussed by the readers of Nanomaterials. From above reason, the manuscript can be published in Nanomaterials.